# Gene Transactivation and Transrepression in MYC-Driven Cancers

**DOI:** 10.3390/ijms22073458

**Published:** 2021-03-27

**Authors:** Marika Scafuro, Lucia Capasso, Vincenzo Carafa, Lucia Altucci, Angela Nebbioso

**Affiliations:** Department of Precision Medicine, Università Degli Studi Della Campania “Luigi Vanvitelli”, Vico Luigi de Crecchio, 7, 80138 Napoli, Italy; marika.scafuro@unicampania.it (M.S.); lucia.capasso@unicampania.it (L.C.); vincenzo.carafa@unicampania.it (V.C.)

**Keywords:** MYC, MYC deregulation, MYC-driven cancers, therapy resistance, therapeutic target, epigenetic modulation

## Abstract

*MYC* is a proto-oncogene regulating a large number of genes involved in a plethora of cellular functions. Its deregulation results in activation of *MYC* gene expression and/or an increase in MYC protein stability. *MYC* overexpression is a hallmark of malignant growth, inducing self-renewal of stem cells and blocking senescence and cell differentiation. This review summarizes the latest advances in our understanding of MYC-mediated molecular mechanisms responsible for its oncogenic activity. Several recent findings indicate that MYC is a regulator of cancer genome and epigenome: MYC modulates expression of target genes in a site-specific manner, by recruiting chromatin remodeling co-factors at promoter regions, and at genome-wide level, by regulating the expression of several epigenetic modifiers that alter the entire chromatin structure. We also discuss novel emerging therapeutic strategies based on both direct modulation of MYC and its epigenetic cofactors.

## 1. Introduction

The *v-myc* oncogene is a transforming factor of the avian virus MC29, observed for the first time in 1964 in chickens affected by spontaneous myelocytomatosis [1,2]. Its human cellular homolog, *MYC*, was identified in Burkitt lymphoma (BL) patients with t(8;14)(q24;q32), in which *MYC* is translocated to the immunoglobulin heavy-chain (IGH) promoter and subsequently overexpressed [3]. MYC is the focus of numerous studies due to its major involvement in many biological processes, such as cell cycle progression, cell growth, apoptosis, differentiation, senescence, self-renewal, pluripotency, and DNA replication. A clear correlation exists between deregulated MYC function and cancer development and progression (Figure 1). In cancer cells, MYC is overexpressed due to aberrations in *MYC* locus, including polymorphisms in *MYC* regulatory sequences, copy number variations, and chromosomal translocations, or by aberrant transduction pathways of *MYC* activation and repression [4]. All these events seem to link *MYC* expression to cancer-associated rearrangements. However, MYC stabilization can also contribute to deregulation in many tumors. This review describes both the pivotal role of MYC in physiology and tumor development, and MYC-targeted anticancer therapies [5,6,7].

## 2. *MYC* Gene Family: Structure and Function

The *MYC* gene family encodes three well-characterized cellular oncogenes, *MYC* (previously known as *c-MYC*), *MYCN*, and *MYCL*, also named “super-transcription factors”, which regulate at least 15% of the human genome. All three oncogenes share significant homology within conserved regions at both terminals. These regions consist of highly conserved elements that provide docking sites for several cofactors that, by regulating both MYC activity and stability, contribute to determining its oncogenic effects. The C-terminal region of these family members includes a common motif consisting of a basic region (b) followed by a helix-loop-helix (HLH), and a leucine-zipper (LZ) domain (b-HLH-LZ motif) [5,6]. HLH and LZ domains mediate protein dimerization, while the adjacent b region promotes binding to DNA-specific sequences (Figure 2) [5]. Through these common motifs, members of the MYC family heterodimerize with MAX, a small b-HLH-LZ factor that binds the b-HLH-LZ of MYC proteins, leading to generation of a functional domain able to bind DNA. Since MYC DNA binding and MYC transactivation are dependent on MAX association, MAX is known as MYC’s obligate partner [6,8,9]. Unlike MAX, MYC cannot homodimerize in vivo [10,11,12]. Protein–protein interaction analysis showed that MAX is able to interact with MYC and MAX itself, even if MYC avidity for MAX leads to an efficient MYC/MAX interaction at equilibrium. Binding assays confirmed that enhancer box (E-box) elements bind preferentially to the MYC:MAX heterodimer rather than the MAX:MAX homodimer [6,8,9]. MAX homodimers do not contain regulatory domains and can recognize the same MYC/MAX motifs, interfering with the action of the heterodimer. Although both MAX:MAX and MYC:MAX complexes bind to the DNA E-box motif CANNTG, they display greater affinity for the canonical MYC E-box sequence CACGTG [6,8,9]. In vitro studies showed that MYC:MAX heterodimers can also bind with low affinity to non-canonical motifs (including proximal E-box variants CATGTG, CACGCG, CACGAG, CATGCG, and CACGTT) and the palindromic hexamer AACGTT. Allevato and colleagues found that high levels of the MYC:MAX heterodimer are required for the low-affinity AACGTT specific binding. This finding was confirmed by in vivo analysis, showing an increase in AACGTT occupancy in the presence of elevated MYC concentrations [9].

The N-terminal of MYC family proteins contains six conserved regions, named MYC homology boxes (MBs). The highly preserved MBs MB0 (amino acids 16–33), MBI (amino acids 45–65), and MBII (amino acids 128–144) are located in the MYC transactivation domain (TAD) (Figure 2). MBs are critical for MYC protein stability, protein–protein interaction, and transcriptional activation or repression of target genes [6,13,14]. MB0 is involved in the interaction with transcription elongation factors and in tumor growth acceleration, while MBI seems to have a role in ubiquitin/proteasome-dependent MYC degradation [6,14]. Multiple ubiquitin ligases are able to control MYC balance. F-box and WD repeat domain-containing 7 (FBW7), for example, was shown to regulate MYC and MYCN stability in response to Ser-62 and Thr-58 phosphorylation within MBI [6]. In particular, Bahram and colleagues showed that mutations abolishing the Thr-58 phosphorylation site can have deleterious effects in BL, increasing the stability and transforming ability of MYC. Specifically, the Thr-58 mutation in lymphoma cells blocks MYC binding to FBW7 and, thus, proteasome-mediated MYC degradation, leading to protein accumulation and tumor progression [15]. Dauch and colleagues identified a further important role for the Thr-58 residue, observing that this threonine is phosphorylated and stabilized by Aurora A protein kinase (AURKA), protecting MYC from proteasomal degradation [16,17].

MBII, the most studied TAD region, seems to play a critical role in several MYC functions such as DNA specific binding, autoregulation, and transforming and transcriptional activities [13]. MBII is able to interact with the transformation/transcription domain-associated protein (TRRAP) [14,18,19] recruiting histone acetyltransferase (HAT) complexes that promote transcription of MYC-bound genes [18].

MBII is also involved in MYC protein stability by binding to S-phase kinase-associated protein 2 (SKP2), a ubiquitin E3 ligase that links MYC transactivation to its ubiquitin-mediated degradation [20]. However, a mutation in MBII can compromise N-terminus integrity, inhibiting most MYC phenotypes and MYC/cofactor binding [21]. Experimental evidence showed that, although with different functions, MB0 and MBII are both involved in MYC-dependent transformation. The proliferation rate and transforming capability of MB0 and MBI deletion mutants were found strongly reduced compared to those of wild-type MYC [14]. In addition, co-expression of non-transforming MB0 and MBII deletion proteins was able to rescue MYC transforming activity, confirming the importance of MB0 and MBII in the tumorigenic function of MYC [14]. The central portion of MYC contains three MBs—MBIIIa, IIIb, and IV—with highly conserved sequences [22] (Figure 2). MBIIIa binds histone deacetylases (HDACs), intervening in MYC-mediated repression; MBIV regulates MYC binding to DNA, but the mechanism is still unclear [21]. MBIIIb binds a hydrophobic cleft of WD-40 repeat protein WDR5 [23]. This interaction is crucial to facilitate MYC recruitment to chromatin regions, regulating the expression of genes linked to protein synthesis [24].

In mutational studies of MYC MBs, Nie and colleagues found that MYC amplification was severely blocked by mutations in MBI and MBII but activated by mutations in MBIII. These findings suggest that each MB region interacts with a specific protein group that has a role in opening chromatin across the transcription cycle [25].

## 3. MYC in Cell Cycle

MYC plays an important role in cell cycle progression, underscored by the correlation between MYC alterations and cell cycle progression impairment [26]. In agreement with the association of MYC overexpression and enhanced cellular proliferation, MYC inactivation or downregulation strongly affects the re-entry of quiescent cells in cell cycle [26]. Several studies showed that the majority of positive cell cycle regulators are MYC targets, including genes encoding for cyclins and cyclin-dependent kinases (CDKs), and for proteins involved in replication [26]. Yap and colleagues showed that MYC overexpression in HOMycER12 cells [27] modulates 37 out of 87 genes involved in cell cycle progression [28]. Specifically, 26/37 genes were regulated to promote progression of the cell cycle: 12 negative effectors were downregulated (including P27KIP1, p18, and GADD45) and 14 positive effectors were upregulated (including CDK4, CDK6, cyclin E1, and CDC25A). Furthermore, Leone and colleagues demonstrated that in order to define cell fate, MYC needs specific E2F activities. In particular, to induce S-phase and apoptosis, MYC requires E2F2/E2F3 and E2F1 involvement, respectively [29,30,31]. However, the precise mechanism underlying MYC-E2F functional interaction is not yet clear [26].

### 3.1. MYC-P27KIP1 Antagonism

The transforming activity of MYC is associated with its ability to stimulate cell cycle progression by promoting G1 to S-phase transition. This progression occurs as a result of several different mechanisms: Upregulation of cyclins (type A-B-D-E1) and CDKs (CDK 1-2-4-6), degradation of cell cycle inhibitors (P27KIP1), and downregulation of CDK inhibitors (p15INK4B and p21CIP1) [32]. MYC antagonistic action against P27KIP1 is recognized as the main mechanism of MYC-driven cancerogenesis [32]. P27KIP1 is degraded when phosphorylated at Thr-187 and, therefore, recognized by SKP2, which targets it for proteosome-dependent degradation. Cyclin E1/CDK2 is considered the main target of P27KIP1 [33,34]. Several reports show that MYC induces cyclin D2/CDK4, causing P27KIP1 sequestration and activation of cyclin E1/CDK2, a key event for MYC mitogenic activity [35,36]. In 2011, Bretones and colleagues observed that in human chronic myeloid leukemia (CML), K562 cells the conditional expression of MYC induced overexpression of SKP2 [37]. MYC was found to bind a region containing E-boxes in SKP2 promoter gene. Furthermore, the expression levels of MYC and SKP2 were positively correlated with each other and negatively correlated with P27KIP1 levels. Importantly, P27KIP1 downregulation was not observed in K562 cells silenced for SPK2, providing strong evidence that SKP2 is a direct target of MYC and its induction was a newly discovered MYC-mediated transformation mechanism [38]. The same research group later found that MYC induces cyclin A and B, which activate CDK1, able to phosphorylate P27KIP1 at Thr-187. In summary, MYC-P27KIP1 antagonism relies on the ability of MYC to induce P27KIP1 degradation by induction of SKP2 and cyclins activating both CDK2 (cyclins A2, E1) and CDK1 (cyclins A2, B1) [32,37].

### 3.2. MYC and MIZ1 Action

MYC-mediated cell proliferation can also result from its effect on the CDK inhibitors p15INK4B and p21CIP1. Although different mechanisms drive inhibition of p15INK4B and p21CIP1 mediated by MYC, the most studied involves MYC-interacting zinc-finger 1 (MIZ1) [26,39]. MIZ1 is a zinc-finger transcription factor that binds to the promoter region of *p15INK4B* gene, leading to overexpression of this inhibitor, which blocks cyclin D-associated kinase activity and arrests cells in G1 phase. When the MYC:MAX heterodimer binds to MIZ1 on the initiator sequence of the *p15INK4B* promoter region, its expression is abrogated, inhibiting cell senescence [40].

MYC binding to MIZ1 is also crucial to inhibit *p21CIP1* transcription. It is well known that p21CIP1 plays an important role in cell growth and differentiation, and that MYC inhibits cellular differentiation. MYC binds to *p21CIP1* core promoter through interaction with the DNA-binding protein MIZ1. In normal hematopoietic differentiation, when MYC levels are low, MIZ1 activates *p21CIP1* expression. With high concentrations of MYC, MIZ1 represses expression of this CDK inhibitor, leading to a block of differentiation [26,41,42]. In addition, by regulating *p21CIP1* expression MYC interferes in p53 response to DNA damage. p53 is able to induce either cell cycle arrest in G1 phase via p21CIP1 activation or apoptosis through induction of *PUMA* and *PIG3*. MYC overexpression does not influence *p53* induction and its ability to bind to the *p21CIP1* promoter region, but does suppress *p21CIP1* transcription activation mediated by p53, sensitizing cells to p53-dependent apoptosis [41].

### 3.3. MYC-p53 Negative Correlation

There is a negative correlation between MYC and p53, considered “yang” and “yin” factors, which act as an accelerator and brake of cell growth, respectively [42,43,44]. Upon stress stimuli, activation of p53 can induce transient cell arrest in the G1 quiescent phase, allowing cells to repair damaged DNA and thus preventing the accumulation of mutations and consequent genomic instability. In the presence of major damage, p53 enhances the transcription of genes that induce prolonged G1 arrest or apoptosis. This ability to eliminate damaged cells makes p53 a master tumor suppressor gene. Since p53 acts as a transcriptional activator of p21CIP1, it is able to induce a blockage of cells in the G1 phase through binding of p21CIP1 to cyclin E1/CDK2 and to cyclin D/CDK4 [45,46]. However, this blockade can be deregulated by proliferation stimuli. Aberrant proliferation without terminal differentiation and DNA damage-induced cell cycle arrest but with high levels of genomic instability can be induced by MYC [47]. In this scenario, it is reasonable to speculate that these two genes affect complementary targets and regulate their activities reciprocally. Previous studies showed a negative correlation between MYC downregulation and p53 activation in control of cell cycle block [48]. Overexpression of p53 reduces MYC levels by interfering with TATA-binding protein (TBP) binding to the TATA motif. In particular, wild-type but not mutated p53 is able to bind TBP, repressing MYC transcription [49]. Ho and colleagues showed that p53 can repress MYC, regardless of p21CIP1 transactivation [46]. In all tested cell lines, mRNA levels of MYC were reduced after γ-irradiation-mediated p53 activation. This downregulation was also found in p21CIP1-deficient cell lines, confirming that MYC repression is not a consequence of p53-dependent p21CIP1 transactivation [50]. In addition, p53-mediated MYC deregulation was abolished by HDAC inhbitor (HDACi) treatment and was accompanied by a decrease in histone H4 acetylation levels at MYC promoter region [46].

### 3.4. MYC-p53 Crosstalk in Tumorigenesis

Several genetic modifications involve *p53* and *MYC*. Most human cancers present such alterations, which drive cancer onset, promotion, and progression as well as therapeutic outcome and resistance [51].

More than 50% of human cancers are characterized by p53 gene mutations, including lung (70%), stomach (60%), colon (60%), esophagus (60%), and aggressive B-cell lymphoma (25%) [52,53,54]. p53 mutations and MYC amplifications frequently co-occur in aggressive tumors [4,55]. Missense point mutations are the most common alterations of p53 and lead to loss of tumor suppressor activity [56]. Nevertheless, p53 mutations in cancer can exert a dominant negative effect on the wild-type p53 allele or gain new oncogenic functions promoting tumor progression and pharmacological resistance [56]. In glioblastoma, Huang and colleagues showed that the phosphatase and tensin homolog (PTEN) exerts oncogenic activity by increasing expression of MYC and Bcl-xL, which is mediated by the gain-of-function mutant p53. Activation of PTEN/mut-p53/MYC/Bcl-xL sustains cell proliferation, colony formation, and invasion [57]. The role of p53 in cancer cell growth has also been extensively investigated in cancer stem cells (CSCs), although the underlying molecular mechanism remains unclear. In breast cancer CSCs, p53 inactivation was associated with constitutive MYC expression, which increases the frequency of symmetric division of CSCs, as revealed by the expression of a large number of mitotic genes useful to identify high-risk patients [58,59]. MYC/p53 crosstalk also plays a pivotal role in B-cell lymphoma (discussed later in a dedicated section) [60].

### 3.5. MYC/BIN1 Interaction: Cell Death Program Regulation

MYC is known to play an important role in apoptosis cell signaling pathways. In the absence of proliferative stimuli, normal cells downregulate *MYC* and exit from cell cycle, while cells with deregulated MYC maintain its expression and die by apoptosis [61,62,63]. MYC-mediated apoptosis ensures correct cell growth in an appropriate cell environment. Tumors overexpressing MYC frequently present mutations that deregulate this cell death program. MBI and MBII are fundamental for MYC transcriptional activation and apoptosis, and present mutations (mainly MBI) responsible for oncogenic activity in many tumors [64,65]. Hence, major scientific efforts have focused on gaining a better understanding of MYC-mediated apoptosis in an attempt to identify key targets inactivated in tumor cells.

Several findings support the important role of the MYC box-dependent-interacting protein 1 (*BIN1*) gene in the MYC-mediated apoptotic program. *BIN1* encodes a nucleocytosolic adapter protein of about 70 kDa and was identified for its ability to inhibit oncogenic activity of MYC by via direct interaction in a MYC domain mutated in cancer [66]. BIN1 expression is ubiquitous in normal cells, while it is often absent in cancer cells. Re-introducing BIN1 causes cancer cells to undergo caspase-independent apoptosis [67]. The tumor suppressor activity of BIN1 was shown to depend on the presence of MBI and MBII [68].

BIN1 is characterized by four main functional regions: An N-terminal BAR region (amino acids 1–249), so called due to its high structural similarity with amphiphysin and RSV167 (BIN1/AMPHIPHYSIN/RVS167); a nuclear localization signal (amino acids 250–256); an MYC-interacting region (amino acids 257–376); and a C-terminal Src homology 3 (SH3) domain (amino acids 377–451). There are two BIN1 isoforms, with or without exon 13, which are ubiquitously and equally expressed. Other isoforms are tissue specific. BIN1 splice variants containing exon 12A abolish the tumor suppressor action of BIN1.

Pineda-Lucena and colleagues clarified the mechanisms underlying binding of BIN1 to MYC and the manner in which BIN1 is regulated by exon 12A [68]. Nuclear magnetic resonance spectroscopy showed that MYC/BIN1 interaction involves the C-terminal SH3 of BIN1 and a consensus class II SH3-binding motif of MYC N-terminal. This short motif is encoded by a proline-rich sequence of MBI and could interact with other SH3 domains. Tumor-specific isoforms of BIN1, including exon 12A, are not able to interact with MYC because an intramolecular interaction occurs between the consensus class I SH3-binding domain, encoded by exon 12, and the SH3 domain, preventing MYC from binding to the SH3 domain of BIN1. Further, MYC phosphorylated at Ser-62, a residue within MBI, is not able to bind BIN1. The negative charge on the side-chain of Ser-62 generates electrostatic repulsion with the acidic BIN1 SH3 domain [68].

MYC/BIN1 interaction also plays a key role in chemoresistance. MYC was shown to increase resistance to cisplatin by downregulating BIN1 [69]. Cisplatin resistance is associated with BIN1 levels, independently of *p53* status. Specifically, BIN1 is able to bind and inhibit poly(ADP-ribose) polymerase 1 (PARP1), abolishing transactivation activity of *MYC*, progression of the cell cycle into M phase, and cisplatin resistance. A model was proposed in which high levels of MYC are able to reactivate PARP1 by repressing *BIN1* via MIZ1, as well as resistance to DNA-damaging agents [69,70]. Wang and colleagues also found a role for BIN1 in inhibiting tumor immune escape in non-small cell lung cancer (NSCLC). In NSCLC, programmed death-ligand 1 (PD-L1) and BIN1 expression are negatively correlated, and PD-L1 expression is inhibited by BIN1. Interestingly, BIN1 overexpression abolishes expression of PD-L1 by inhibiting MYC and EGFR/MAPK signaling pathways. These findings open the way for more effective immunotherapies for NSCLC treatment based on reactivation of BIN1, which is able to reverse immune escape mediated by PD-L1 by inactivating MYC and EGFR/MAPK networks [71].

## 4. Role of MYC in the Homeostasis of Hematopoietic Stem Cells

Hematopoietic stem cells (HSCs) are able to produce all types of mature differentiated blood cells and, simultaneously, renew themselves via an as yet undefined mechanism. The life span of differentiated erythrocytes in blood is limited: They are periodically released into circulation to replace cells lost due to normal tissue turnover. In normal conditions, this assures the homeostatic balance between HSCs and differentiated cells, which is adaptable to physiological needs. This steady-state dynamic equilibrium is reached through crosstalk between HSCs and their microenvironment, the so-called stem cell niche [72]. MYC controls hematopoietic cell proliferation and the balance between HSC self-renewal and differentiation by modulating HSC adhesion to the niche and/or migration [73]. In the niche, quiescent HSCs expressing low levels of MYC are connected to both spindle-shaped N-cadherin^+^ osteoblasts, embedded in stromal fibroblasts, by homotypic N-cadherin and LFA-1/ICAM interaction, and to the specialized extracellular matrix by integrin bindings. Upon mitogenic stimuli, HSCs enter the cell cycle, creating two daughter cells, one of which differentiates and the other remains in the niche. If MYC is not activated, cell-adhesion molecules continue to be highly expressed in the two daughter cells, which remain in the niche and do not differentiate. MYC induction in only one of these cells reduces cell adhesion molecule expression, creating asymmetry and ensuring homeostasis: One of the daughter cells remains in the niche while the other exits, promoting differentiation. High expression levels of MYC in both daughter cells causes a departure of the cells from the niche, leading over time to depletion of the HSC pool.

In homeostatic conditions, MYC expression levels are low in a subset of long-term HSCs that maintain self-renewal capacity and high in another subset of long-term HSCs that differentiate into short-term HSCs. During differentiation, MYC levels increase, cells lose their self-renewal capacity, and early progenitor cells become transient-amplifying cells that are less and less multipotent but with high MYC levels to ensure an active cell cycle. MYC expression is downregulated at the end of differentiation, leading to cell cycle exit. Together, homeostasis of HSCs is controlled by MYC in terms of both self-renewal and differentiation [73,74].

### 4.1. Alterations of MYC Pathways in Lymphoma and Leukemia

Most malignant lymphomas are mature B-cell neoplasms. These diseases arise from neoplastic transformation of normal B-cells at different stages of lymphomagenesis. MYC aberrations (amplifications or translocations) have been identified in most types of B lymphoma. These alterations result in MYC overexpression [60].

#### 4.1.1. MYC in Acute Myeloid Leukemia

In 1986, MYC deregulation was first described in erythroid differentation of Friend murine erythroleukemia cells [75]. However, scientific interest in this transforming activity has been increasing since new MYC-targeting drugs showed clinical activity in acute myeloid leukemia (AML) [76]. These compounds were either direct (e.g., 10058-F4) [77] or indirect (e.g., bromodomain and extra-terminal motif inhibitors [BETi]) inhibitors of MYC activity. Frequent aberrations that upregulate MYC in AML are FMS-like tyrosine kinase internal tandem duplications [78,79].

In leukemiogenesis, aberrant expression of MYC directly modulates expression of EZH2 and miR-26a. EZH2 is a member of polycomb repressive complex 2 (PRC2) that determines aberrant transcriptional gene silencing observed in AML [80,81]. During myeloid differentiation in AML, miR-26a is activated while MYC and EZH2 are downregulated. In AML, EZH2 is transcriptionally modulated by MYC. Induction of miR-26a (i) leads to cyclin E2 suppression responsible for blocking cell cycle progression, (ii) reinforces antiproliferative effects of vitamin D, and (iii) triggers cell differentiation [81].

MYC also appears mutated in myelodysplastic syndromes and myeloproliferative neoplasms [82].

#### 4.1.2. MYC in Double-Hit Lymphoma

Double-hit lymphoma (DHL) is an aggressive type of B-cell non-Hodgkin lymphoma characterized by genetic translocations involving *MYC* and either *BCL2* or *BCL6*, even if, rarely, all three rearrangements can occur together (triple-hit lymphoma) [83,84,85]. As regards gene mutations, DHL shares features with diffuse large B-cell lymphoma (DLBCL) and BL [84]. MYC and BCL2/BCL6 are upregulated and are able to act in combination. Prognosis of DHL patients is poor. Li and colleagues tested BETi action on MYC activity and found that in both MYC/BCL2 DHL and MYC/BCL2/BCL6, triple-hit lymphoma cells were responsive to BETi (JQ1, I-BET, OTX) treatment [86]. However, although BETi treatment is able to downregulate MYC and BCL6 expression, it has no effect on BCL2.

#### 4.1.3. MYC in Chronic Myeloid Leukemia

CML is a clonal myeloproliferative disease characterized by t(9;22)(q34;q11) reciprocal translocation. This translocation, also known as Philadelphia chromosome, gives rise to a hybrid protein (BCR-ABL1) [87]. BCR-ABL1 is a protein-tyrosine kinase whose constitutive expression is under control of the BCR promoter and is responsible for leukemic transformation and progression from chronic phase to blast crisis [88,89]. Since the causative molecular event is unique—generation of *BCR*-*ABL1* oncogene—drugs targeting the tyrosine kinase activity of BCR-ABL1, such as imatinib mesylate, have proved highly effective as anti-CML therapies [88].

BCR-ABL1 is able to regulate MYC expression through PI3K and JAK2 pathways as well as the E2F1 transcription factor, and to activate JAK2, thereby inhibiting MYC proteasome-dependent degradation [89,90]. Sharma and colleagues studied BCR and BCR-ABL1 expression levels in conditions of MYC overexpression and silencing [89]. In CML MYC-overexpressing cells, the MYC:MAX heterodimer was found to bind to BCR promoter region in four binding sites, leading to upregulation of both BCR and the fusion protein at mRNA and protein level. Conversely, when MYC was silenced in CML cells, BCR and BCR-ABL1 expression was downregulated, causing a block in cell proliferation and induction of cell death. These findings confirm the important role of MYC in *BCR* promoter regulation. A positive feedback mechanism exists between MYC and BCR-ABL1: The fusion protein increases MYC activity and MYC induces its transcription, binding directly to the *BCR* promoter region. Given that MYC is upregulated during blast crisis, transcriptional regulation by MYC of *BCR* promoter may be a crucial event in upregulation of the fusion protein and thus in the aggressiveness of CML [88,89].

#### 4.1.4. MYC in Burkitt Lymphoma

BL is the disease model caused by MYC deregulation in which the t(8;14) translocation juxtaposes the IGH locus with *MYC* [91]. Following this translocation, *MYC* is transcriptionally controlled by immunoglobin enhancer elements and constitutively expressed [92]. In addition to chromosome 14, found in 10% of BL cases, *MYC* can translocate to chromosome 2 or 22, juxtaposing with kappa or lambda light chain genes, respectively [92]. BL is an uncommon form of highly aggressive B-cell non-Hodgkin lymphoma, in whose pathogenesis several other oncogenes, such as *TCF3*, collaborate with MYC [91]. High MYC levels negatively affect HLA class II-mediated antigen presentation, which contributes to the immuno-evasive characteristics of BL cells [93]. Constitutive expression of *MYC* exerts pleiotropic effects on a large group of genes, making MYC and its signaling pathways an important target for therapeutic strategies.

Heat shock protein (HSP) 90 is important for the correct folding and activity of regulatory proteins involved in tumorigenesis, which make cancer cells “addicted” to HSP90 [94]. HSP90 inhibitors, used for the treatment of different cancer types, competitively bind the N-terminal ATP-binding site of HSP90 leading to client protein degradation. Of note, MYC was found to be a client protein of HSP90, and HSP90 inhibitors are able to downregulate the MYC transcriptional program, leading to growth arrest and apoptosis [95]. Moreover, HSP90 expression is elevated in BL cells, indicating its key role in MYC programs [96]. In line with these findings, Poole and colleagues demonstrated that in BL cells inhibition of HSP90 is able to reduce MYC levels by blocking its transcription and increasing its degradation, suggesting that HSP90 inhibitors may provide an alternative approach in BL treatment [97]. Targeting the MYC/HSP90 axis has thus proved to be a promising strategy to regulate the function/stability of MYC.

## 5. MYC-Mediated Transcriptional Output Regulation

Although the role of MYC as a driver in many human cancers is clear, the underlying molecular mechanisms still need to be elucidated and the possibility of selective targeted therapies remains in doubt. MYC exerts its oncogenic action by controlling the expression of a panel of genes: Some genes are activated, and others are silenced. Both the activating and repressing functions of MYC are fundamental to its oncogenic activity and are mediated by binding to other transcription factors [98,99] and by recruitment of chromatin remodeling factors in binding sites [97]. This would explain the frequent non-predictivity of MYC genomic localization profiles as well as the slight overlap of MYC target gene signatures found in different cells [98,99].

MYC selectivity on transcriptional programs is the subject of intense debate: MYC has been proposed as both an amplifier [100,101] and a selective regulator [102,103,104] of transcriptional cellular programs.

In an attempt to elucidate how MYC recognizes its DNA binding sites, Guccione and colleagues found that these sites displayed distinctive histone marks, identifying two main promoter populations [105] that coincided with previously described high- and low-affinity MYC binding sites [99]. One population, with high affinity for MYC, was enriched for active histone marks (H3K4me2/3, H3K79me2, and H3acetyl-lysines), whereas the other was associated with inactive histone marks (H3K27me3) [105]. Since most H3K4me3 regions overlap with CpG islands they are called “euchromatic islands” and are crucial for MYC binding in vivo [105]. MYC binding to DNA occurs after recognition of these chromatin marks, regardless of the presence of an E-box sequence. E-boxes outside euchromatic islands were found not significantly bound to MYC, whereas those within islands were virtually all bound. The recognition of E-boxes therefore remains important [106], but secondary to euchromatic islands [105]. These observations are consistent with the idea that histone marks are read by specialized proteins that bind MYC, which, once bound to DNA, induces local hyperacetylation of histones H3 and H4 [107]. MYC is physiologically able to bind thousands of promoters [108] acting as a universal transcription amplifier. Nie and colleagues demonstrated that in primary activating lymphocytes, MYC binding occurs on promoters located within open chromatin and is weakly dependent on E-boxes [100,105]. In addition, MYC binding is related to the amount of RNA pol II preloaded on these promoters and activates transcription through release of paused RNA pol II [100]. Hence, in homeostatic conditions, physiological levels of MYC may trigger an immediate response to mitogenic growth factors that lead to an accumulation of macromolecules required for proliferation and other preprogrammed pathways [22,108,109] until other regulatory networks intervene. In cancer, prolonged overexpression of MYC could lead to chronic deregulation involving all cellular gene expression programs, which become independent of growth stimuli, causing uncontrolled proliferation (Figure 1) [101]. When MYC levels are increased in pathological conditions, enhancers are occupied and many cell networks are deregulated [101]. Lin et al. reported that different MYC levels could differentially regulate gene expression. In cancer cells with low MYC levels, MYC co-occupies with RNA pol II promotor regions of actively transcribed genes [101,105,110,111,112]. In tumor cells overexpressing MYC, increased MYC levels lead to saturation of the same promoters but also to enhancers of active genes (a phenomenon known as “invasion”), as reported by other previous studies [113,114,115]. Specifically, in the presence of elevated MYC, the excess of MYC binds lower-affinity E-box sequences in core promoters and enhancers of active genes, with insignificant binding to new gene regions [101]. Genes with elevated MYC occupancy at enhancers showed the greatest increase in RNA pol II elongation, leading to amplified transcription of existing programs of transcription [101]. In line with these findings, Lorenzin and colleagues correlated promoter occupancy to gene expression with different levels of MYC, reporting that specific gene expression profile variations in cancer cells are due to increased MYC. Tumoral levels of MYC lead to its stronger interaction with binding sites of genes that in normal conditions weakly bind MYC and that codify for G-protein coupled receptors and for proteins engaged in nutrient transport and hypoxic response. Genes already strongly bound to MYC in normal cells, associated for example with ribosome function, were not modulated by increased tumor-specific MYC levels [116]. The authors also found that MYC:MAX heterodimer occupancy was dependent on its interaction with resident chromatin proteins such as WDR5 [23] and RNA pol II transcriptional machinery, supporting MYC’s amplifier function [116].

In order to discriminate between direct and indirect effects of MYC on transcription, Sabò and colleagues compared genome-wide ChIP and transcriptome profiles of actively proliferating and quiescent cells. They found that RNA amplification and invasion of promoters and enhancers are independent phenomena of MYC overexpression. Although MYC can interact with almost all regulatory elements of the genome, it does not directly act as a global transcriptional amplifier: RNA amplification is an indirect effect of MYC that drives, regardless of amplification, the differential expression of specific groups of target genes, leading to changes in the state of the cell, which can in turn affect total RNA production and turnover [103]. Tumor cells differ from normal cells in the expression of a specific set of genes that are up- and downregulated by different levels of MYC. Walz and colleagues described three factors that may explain this phenomenon: (i) Promoter binding affinity: Physiological and elevated levels of MYC regulate functionally distinct classes of promoters that differ in E-box sequence affinity to MYC; (ii) transcription initiation: MYC positively and negatively influences the initiation of transcription regardless of transcriptional elongation, recruiting TFIIH and P-TEFb/Cdk9, important factors for RNA pol II phosphorylation [105,111,117,118]; (iii) MYC/MIZ1 ratio: MYC/MIZ1 complex suppresses multiple MYC target genes and MYC/MIZ1 ratio regulates cell response to MYC. Genes with a lower MYC/MIZ1 ratio are regulated by MIZ1 and its binding sequences; MYC-repressed promoters harbor E-box sequences able to bind SP1, which in turn binds both MYC and MIZ1, suggesting that protein–protein interactions influence the ratio at each promoter. Experiments with shMIZ1 showed that MIZ1 is required for the repression of a large number of, but not all, MYC target genes [104]. Baluapuri and colleagues showed that MYC levels can influence RNA pol II association with transcription elongation factors including SPT5. Specifically, MYC directly recruits SPT5 at RNA pol II in a CDK7-dependent manner, consequently potentiating the processivity and elongation rate of RNA pol II. Enhanced MYC levels in tumors induce the “squelching” of SPT5, subtracting this and other components of elongation machinery from genes that are known targets of MYC-dependent repression, such as those encoding TGFβ pathway proteins and regulators of immune system in cancer. It is therefore plausible to think that tumors exploit SPT5 squelching to repress tumor suppressor genes, leading to aberrant cell growth [119].

To distinguish between direct and indirect effects mediated by MYC, it is crucial to identify DNA-binding events. Muhar and colleagues provided an insight into the regulatory roles of MYC and the BET family protein BRD4 [120]. By analyzing transcriptional profiles following acute degradation of BRD4 or MYC, they found that: (i) BRD4 degradation induced a global downregulation of transcription, suggesting that BRD4 is a general coactivator of RNA pol II; (ii) the BETi JQ1 affected transcription output in a dose-dependent manner, with high doses suppressing global transcription output and low doses affecting only JQ1-sensitive genes, such as MYC. Sensitivity to JQ1 was also related to the presence of other factors in JQ1-sensitive genes, indicating that other proteins mediate BRD4 recruitment/activity on chromatin; (iii) MYC degradation resulted in an unaffected transcriptome with only a few hundred genes repressed. MYC-dependent genes are involved in the synthesis of proteins and nucleic acids that in turn produce secondary effects, such as RNA synthesis, which contribute to malignant transformation [102]. These findings support the role of MYC as a specific regulator of transcriptional output [120,121].

This new paradigm for MYC function has paved the way for the development of novel anticancer therapies (as discussed in Section 6).


### 5.1. MYC-Dependent Transactivation

MYC-dependent transactivation of genes containing a canonical E-box is the best-known mechanism by which the MYC:MAX heterodimer activates its target genes through recruitment of the adaptor protein TRRAP [18,122]. TRRAP acts as a scaffold to recruit HAT-containing complexes such as Spt-Ada-Gcn5-Acetyl transferase (SAGA) complex or Tip60 HAT-containing complex as well as the SWI/SNF-related histone exchange protein and p400, which binds to MYC [18,122,123,124]. This assembly, with GCN5 acetylating H3 at K9, K14, and K18, and TIP60 acetylating H4 at K5, K8, and K12 as well as H2A at K5, leads to a state of hyperacetylation that makes chromatin accessible to RNA pol II and activates transcription of target genes (Figure 3A) [125].

Via another mechanism, MYC recruits the co-factors p300/CBP, which play an important role in both the transactivating function and stability of MYC [126]. p300 and CBP activate transcription by increasing histone acetylation and open chromatin to transcriptional machinery. Here, recruitment is again mediated by TRRAP. However, Faiola and colleagues demonstrated that p300 directly binds MYC in its N-terminal TAD region and causes its acetylation at six lysine residues within the TAD and DNA-binding domain [127].

p300/CBP is thought to have a dual effect on MYC: Stabilizing MYC independently of acetylation, acting as a MYC coactivator, and inducing proteosome-dependent degradation of MYC via its acetylation, acting as an inducer of MYC instability [126]. In support of this hypothesis, depletion of p300 in CBP-knockout and -deficient cells leads to downregulation of MYC and cell death. Specifically, p300 depletion determines aberrant chromatin modifications within *MYC* gene locus, decreasing the occupancy of acetylated H3K18 and acetylated H3K27, and the recruitment and phosphorylation of RNA pol II [128]. These findings suggest that patients with cancers overexpressing MYC could benefit from therapy using p300 inhibitors. 

MYC is also a substrate of two other HATs, GCN5/PCAF and TIP60, which by acetylating MYC cause increased stability of MYC protein [129]. This indicates that HAT enzymes play a different role in regulating MYC functions and that a bidirectional regulation exists between these cofactors and MYC.

In addition to HATs, lysine-specific histone demethylases (KDMs) are recruited by MYC to trigger transcription of target genes. A growing body of evidence shows that MYC directly binds with KDM4B and recruits KDMs on the E-box of target genes [130,131], suggesting that MYC is able to modulate chromatin methylation status by decreasing inactive marks such as H3K9 methylation. Targeting KDM4B function could represent a promising tool to downregulate MYC and induce cell cycle block.

Other co-factors recruited by MYC at E-box target genes include two protein kinases, proviral integration site (PIM) 1 and 2. PIM kinases are involved in the tumorigenic action of MYC by increasing its stability or modulating chromatin structure. Upon growth stimuli, PIM1 binds to MBII and phosphorylates histone H3 at Ser-10, which recruits RNA pol II, participating in MYC-mediated transactivation of target genes [132,133]. PIM1 phosphorylates MYC at Ser-62, while PIM2 phosphorylates MYC at Ser-329, increasing its stability in both cases [134].

### 5.2. MYC-Dependent Transrepression

MYC-driven functions in tumor initiation and progression also depend on transcriptional repression of target genes and recruitment of chromatin modifying factors [104,135].

The best-characterized transrepression is mediated by the MYC/MIZ1 interaction. The MYC/MIZ1 complex is able to recruit the DNA methyltransferase 3A (DNMT3A) corepressor to the promoter region of their target genes, *p21CIP1CIP1* and *p15INK4B* [136]. This ternary complex, MYC/MIZ1/DNMT3A, methylates and actively suppresses *p21CIP1* and *p15INK4B* promoter activity, leading to cell proliferation rather than differentiation and senescence during tumor progression. In NSCLC, silencing of the Ras association domain-containing protein tumor suppressor gene requires recruitment of DNMT3A by MYC (Figure 3B) [137].

Two HDACs, HDAC1 and HDAC3, are reported to interact with MYC [138,139]. MYC recruits HDAC1 on the promoter region of tissue transglutaminase (tTg) [139] and HDAC3 on the promoter of hyperplastic polyposis protein 1 (HPP1) [140], resulting in histone hypoacetylation and transcriptional suppression. MYC also represses microRNA expression by recruiting HDAC3 [141], suggesting that HDAC recruitment may be a general MYC mechanism of action. 

Other important factors involved in MYC-mediated transrepression are Pontin (TIP49) and Reptin (TIP48), two transcriptional cofactors with ATPase and helicase activities [142]. Affinity chromatography analysis showed that these corepressors are MYC N-terminal interacting proteins [143]. Since both proteins interact with the MBII domain, it is plausible to think that they are important for MYC transforming activity [142].

### 5.3. BPTF: MYC Co-Factor for Chromatin Remodeling in Human Cancer

The mechanisms underlying MYC-mediated chromatin recognition are not well understood. It has been hypothesized that MYC accessibility to DNA involves an ATP-dependent nucleosome remodeling enzyme, named NURF. This enzyme uses ATP hydrolysis to catalyze correct and adequate nucleosome sliding [144]. MYC interacts with the bromodomain PHD finger transcription factor (BPTF), the largest of the three NURF subunits (the others being SNF2L and pRBAP46/48). BPTF interacts with transcription factors, histone variants, and modified histones (H3K4me3, H2A.Z, H4K16ac), giving sequence specificity to NURF [144,145,146]. Clinical studies showed that BPTF is mutated in several pathologies such as syndromic neurodevelopmental anomalies, lung cancer, and bladder tumors [146]. It has been postulated that the effects of BPTF might be mediated by MYC. Bioinformatics analyses highlighted a positive correlation between MYC and BPTF in MYC-addicted tumors including BL, as well as prostate, colorectal, and pancreatic cancer [146]. Richard and colleagues found that BPTF is necessary for MYC transcriptional activity. BPFT silencing prevents interaction with MYC and robustly affects its tumorigenic activity [146]. Specifically, in absence of BPTF, DNA accessibility in target genes is generally low and MYC associates with low-affinity promoters, strongly affecting cell proliferation and delaying tumor development. These findings suggest that BPTF inhibition may be a valuable tool for the treatment of MYC-addicted cancers [146].

## 6. Therapeutic Strategies to Target MYC

The oncoprotein MYC is found deregulated in many human cancers. A greater insight into key functions mediated by MYC in tumor environment is crucial for the development of more effective therapeutic approaches. The abnormal activation of MYC in cancer results from transcriptional overexpression and/or protein stabilization. Growth arrest, apoptosis, and differentiation occur upon a reduction in MYC, making MYC an attractive target for anticancer therapy. Since MYC lacks enzymatic activity and deep targetable pockets, it is not a druggable protein. Diverse therapeutic strategies have been proposed to target MYC but are still considered an elusive goal. However, inhibiting MYC co-factors and/or activating MYC repressors could represent a valid strategy to bypass this limit (Figure 4).

### 6.1. Targeting Epigenetic Mechanisms Controlled by MYC

To overcome the lack of globular functional domains within MYC, a growing number of pharmacological approaches aim to target epigenetic modifications to correct the epigenome in MYC-addicted cancer. The possibility of using inhibitors of DNMTs, HATs, HDACs, histone methyltransferases (HMTs), KDMs, and BETs has been extensively investigated (Figure 4).

Two DNMT inhibitors (DNMTi), decitabine and azacitidine, were shown to be effective in inhibiting the tumorigenic effects of translocated MYC in BL, in some forms of DLBCL, and in other types of non-Hodgkin lymphoma [147,148]. However, the use of DNMTi requires careful evaluation as at high doses these compounds are very cytotoxic, causing off-targets effects, and pan-inhibition of DNMTs could itself promote tumorigenesis. Exploiting selective DNMTi rather than broad demethylating agents could thus prove to be a more promising therapeutic strategy [149].

Histone acetylation is the main mechanism by which MYC activates transcription of its target genes. As widely discussed, HATs are both MYC co-factors and regulators of its turnover. HAT inhibitors (HATi) therefore represent attractive weapons against MYC-driven cancers. In recent years, a considerable number of HATi have been discovered. PU139 and PU141 both block neuroblastoma cell growth in vivo, PU139 inhibits the activity of GCN5, PCAF, CREB, and p300/CBP, and PU141 only that of p300/CBP [150]. Recently, TIP60 inhibitors have been emerging as promising therapeutics, but further investigations are required [151].

The role of p300/CBP is crucial for the transactivating activity of MYC. Although several inhibitors have been identified, C646 remains the most potent and selective. This molecule is able to downregulate MYC expression in CBP-deficient cancers, promoting apoptosis [128]. As with DNMTi, p300 loss can accelerate leukemogenesis in mice with myelodysplastic syndrome [152]. Further studies are needed in order to translate the use of HATi into clinical practice. 

HDACs have been actively investigated for their ability to unlock the MYC-mediated repression of tumor suppressor genes. HDACi (such as SAHA) were recently shown to induce acetylation of MYC at K323, resulting in MYC downregulation at both mRNA and protein level, thus releasing expression of target genes, including *TRAIL*, and leading to cancer cell apoptosis [153]. Inhibition of HDAC2 by SAHA was able to downregulate MYC and to reactivate *TBP-2* expression, blocking cell growth and inducing apoptosis in many cancer cell lines [154]. In neuroblastoma and breast cancer, tTG expression is reactivated by HDACi, which prevent HDAC1 recruitment by MYC [139]. A number of HDACi are currently in clinical trials [155] and are yielding exciting results in MYC-driven cancers. 

Several molecules with HMT and KDM inhibitory activity have been proposed. GSK126, a selective inhibitor of EZH2 HMT action, is able to induce apoptosis in multiple myeloma by reducing H3K27me3 [156]. KDM1A inhibition by HCI-2509 downregulates MYC expression in prostate cancer cells, suggesting that it may act as a promising therapeutic in castration- and docetaxel-resistant prostate cancer [157]. Ciclopirox targets KDM4B, inhibiting MYC signaling pathways and tumor growth in MYC-driven neuroblastomas [158]. SD70, a KDM4C-selective inhibitor, represses MYC transcription and activates p53, inducing apoptosis in glioblastoma [159]. 

Ever since the role of super-enhancers was identified in tumorigenesis, scientific efforts have focused on discovering inhibitors of BET. By targeting MYC, JQ1 showed anticancer activity in several cancer types including Merkel cell carcinoma [160], esophageal squamous cell carcinoma [161], and EBV-positive nasopharyngeal carcinoma [162], as well as in a variety of leukemias and lymphomas [163,164,165]. In combination with other epigenetic drugs, BETi have also shown promising therapeutic effects. In lymphoma cells treated with BETi plus HDACi, the BETi RVX2135 sensitizes MYC-overexpressing lymphoma cells, triggering the re-expression of HDAC-silenced genes [166]. In AML cells, JQ1 synergizes with the SUV39H1 inhibitor chaetocin, increasing antiproliferative and differentiative effects [167].

### 6.2. Inhibitors of MYC:MAX Heterodimerization

The inability of MYC to heterodimerize with its obligate partner MAX led to the development of different antagonists of MYC/MAX dimerization, with the aim of blocking MYC oncogenic activity (Figure 4) [168,169]. Due to their low target specificity, fast metabolism, and low potency, no MYC/MAX inhibitors have as yet entered clinical trials. Nevertheless, a number of small molecules that interfere directly with MYC/MAX or MYC/DNA binding have been identified. The first molecules to be described were IIA6B17 [170] and NY2267 [171], able to block MYC-mediated malignant transformation by inhibiting MYC/MAX dimerization. These molecules later showed low specificity, inhibiting other proteins containing LZ domains such as JUN [172]. Their lack of specificity has so far limited their use. 

The compounds 10058-F4 and 10074-G5 were discovered in 2003. These molecules bind MYC and prevent it from acquiring the correct conformation to complex with MAX [11]. Several studies were performed to enhance the potency of these compounds [173], which show low potency in vivo despite good in vitro efficacy [174,175].

Promising in vivo results were also obtained using Mycro3 and MYCMI-6. Mycro3 is derived from pyrazolopyrimidines acting against MYC/MAX dimerization [176]. In MYC-dependent pancreatic ductal adenocarcinoma, oral administration of Mycro3 was able to induce cell growth arrest and apoptosis [177]. MYCMI-6 is a strongly selective MYC /MAX inhibitor, and inhibited cell growth and induced apoptosis in MYC-driven xenograft models. No effect on MYC expression was observed upon challenging with MYCMI-6, which specifically targets MYC/MAX [178].

Recently, Han and colleagues identified two small molecules, MYCi361 and MYCi975, showing promising anticancer activity with a positive pharmacokinetic profile, as indicated by high plasma concentrations, increased half-lives, and tumor penetration. In several cancer cell lines, these compounds downregulated MYC expression and decreased MYC stability by increasing its phosphorylation at Thr-58, driving it to proteosome-dependent degradation. This action led to the suppression of MYC-dependent cancer cell proliferation and tumorigenicity both in vitro and in vivo. Interestingly, MYCi361 and MYCi975 are also able to affect tumor immune microenvironment, increasing PD-L1 expression [179]. 

Other studies have focused on identifying small molecules able to inhibit the binding of MYC to E-box DNA sequences. Celastrol is a natural compound that inhibits DNA binding by altering the structure of the MYC:MAX heterodimer without causing its dissociation [180]. 

The best-known peptide-based MYC inhibitor is Omomyc [181]. Omomyc is the MYC b-HLH-LZ domain containing four amino acid substitutions (E410T, E417I, R423Q, R424N). These substitutions eliminate electrostatic clashes, which inhibit MYC dimerization but not its homodimerization or heterodimerization with MAX. This peptide is able to impair clonogenicity and cell growth. Three mechanisms of actions have been hypothesized: (i) Omomyc dimerizes with wild-type MYC or homodimerizes, blocking MYC:MAX heterodimer formation and its binding to E-box DNA sequences; (ii) Omomyc heterodimerizes with MAX, sequestering MAX from MYC; (iii) Omomyc induces proteosome-dependent MYC degradation (a mechanism dependent on (ii)). Omomyc is a useful tool to better understand the functional effects of MYC signaling pathway interference.

## 7. Concluding Remarks

MYC deregulation is a signature of many types of human cancers. MYC overexpression causes tumorigenesis by deregulating genetic and epigenetic checkpoint mechanisms which control cell proliferation, apoptosis, and differentiation. Cells that overexpress MYC bypass these mechanisms, acquiring many “hallmarks” of cancer, most notably unlimited and uncontrolled cell proliferation. MYC activation is a sign of malignant growth, induces self-renewal of stem cells, and blocks senescence and cell differentiation. MYC also regulates tumor microenvironment through activation of angiogenesis and suppression of the host immune response. Several studies indicate that MYC is a regulator of cancer genome and epigenome: MYC modulates expression of target genes in a site-specific manner, by recruiting chromatin remodeling co-factors at promoter regions, and at genome-wide level, by regulating the expression of a plethora of epigenetic modifiers that alter the entire chromatin structure. By exploiting this mode of regulation, novel therapeutic strategies have emerged based on both direct modulation of MYC and its epigenetic cofactors.

Inhibiting MYC at physiological levels, both directly and indirectly, is able to restore mechanisms regulating key checkpoints in cell cycle progression, leading to tumor regression, senescence, and apoptosis. By eliciting immunoresponse and blocking angiogenesis, MYC inhibition can also modify the tumor microenvironment. Indirect MYC-targeted therapies are currently very appealing as they are based on the reversibility of epigenetic modifications.

Despite huge efforts, the mechanisms underlying MYC-driven tumors still remain unclear and no MYC-targeted therapies are currently available in the clinic. Nevertheless, drug-discovery studies have enhanced our understanding of the stumbling blocks that limit MYC targeting. Indeed, over the years, new approaches have been identified to develop potent molecules with positive pharmacokinetic profiles, and new mechanistic insights have been gained, thus paving the way toward the development of a truly effective inhibitor in MYC-driven cancers.

## Figures and Tables

**Figure 1 ijms-22-03458-f001:**
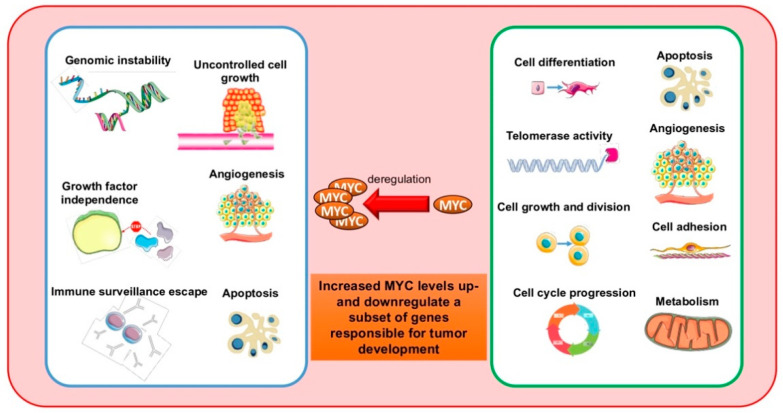
MYC regulation of distinct cellular functions. Many biological processes are regulated by MYC. Elevated MYC levels can result in deregulation of a subset of genes involved in cancer development.

**Figure 2 ijms-22-03458-f002:**
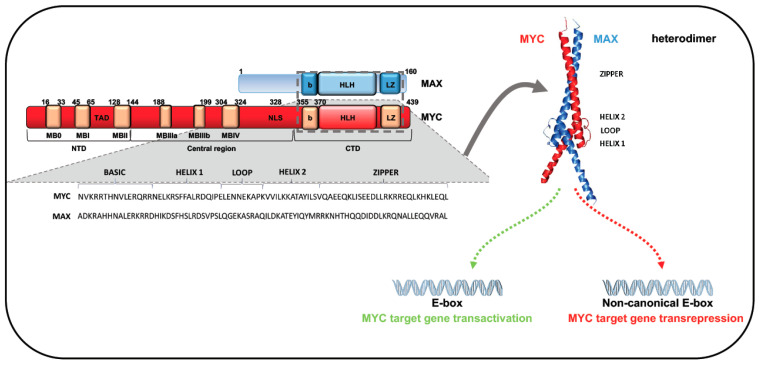
Domain architecture of MYC protein and its interacting partner MAX, showing MYC protein residues and their different functional characteristics. N-terminal region interacts with several partners and C-terminal region binds MAX via b-HLH-LZ motif. MYC:MAX heterodimer binds canonical and non-canonical E-box DNA sequences of target genes.

**Figure 3 ijms-22-03458-f003:**
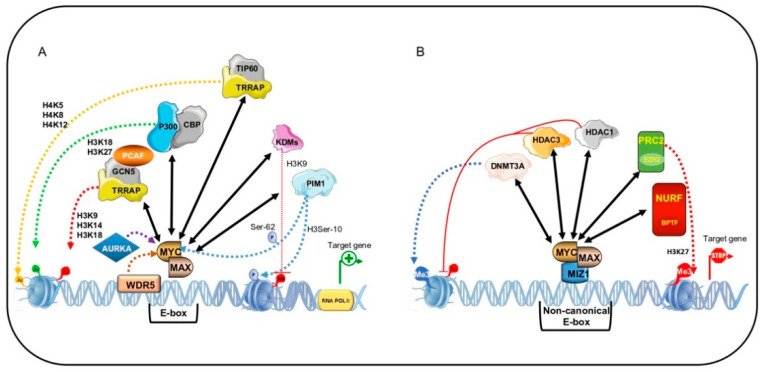
MYC-mediated transactivation and transrepression of target genes. Transcriptional activation of target genes (**A**). MYC:MAX heterodimer binds E-box sequences and transactivates canonical target genes through recruitment of chromatin modifying co-factors. TIP60 and GCN5 via transformation/transcription domain-associated protein (TRRAP) and p300/CBP, respectively, increase acetylation of histone H3 and H4, inducing an open chromatin conformation allowing RNA polymerase II machinery to bind the core promoter. Histone demethylases (KDMs) regulate the methylation of histone H3 by removing repressive chromatin marks, thereby contributing to gene activation. PIM1 phosphorylates histone H3 and MYC increasing its stability. Aurora A protein kinase (AURKA) phosphorylates and stabilizes MYC. WDR5 is crucial for recruitment of MYC at chromatin regions. Transcriptional repression of target genes (**B**). MYC:MAX/MIZ1-mediated transrepression of non-canonical target genes involving recruitment of chromatin co-repressors. MYC:MAX/MIZ-1 complexes recruit DNMT3A to non-canonical targets, thereby increasing tri-methylation of histone H3, promoting tumor cell proliferation rather than differentiation and senescence. HDAC1 and HDAC3 contribute to histone deacetylation and thus gene silencing. EZH2 is a member of PRC2 that catalyzes tri-methylation of histone H3 to enhance gene silencing. Bromodomain PHD finger transcription factor (BPTF) is necessary for MYC transcriptional activity.

**Figure 4 ijms-22-03458-f004:**
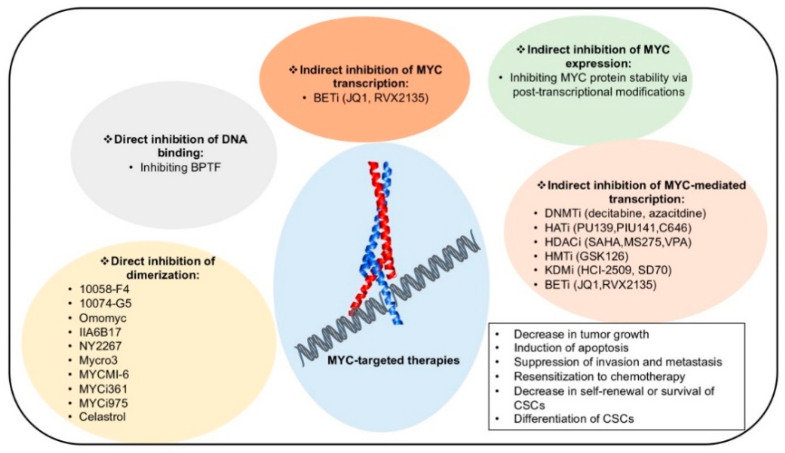
MYC-targeted therapies. Current therapies targeting MYC using direct and indirect inhibition strategies leading to tumor suppression.

## Data Availability

Not applicable.

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
