# Peer review of "Gene Transactivation and Transrepression in MYC-Driven Cancers"

_ijms, 2021, doi:10.3390/ijms22073458_

Round 1

Reviewer 1 Report

The work improved and I can recomand publication in IJMS.

Author Response

Thank you very much for your comments.

Reviewer 2 Report

The MYC oncogene contributes to the genesis of many human cancers. MYC lies at the crossroads of many tumor-related growth-promoting signal transduction pathways, it has a central role in almost every aspect of tumor progression. The MYC gene itself is under tight transcriptional control, its deregulation is a hallmark of malignant growth, and is frequently associated with poor prognosis and low survival rate. In this review, Scafuro et al summarized the latest advances in the understanding of MYC. The authors reviewed the structure and function of the MYC gene family, the role of MYC in growth-related signaling pathways, homeostasis of hematopoietic stem cells, MYC-regulated transcriptional pathways. In the end, the authors also summarized the current strategies to target MYC for therapeutic, both through direct modulation of MYC itself and its epigenetic cofactors, as MYC is at the crossroads of many important biological pathways.

Overall, this review is well organized and comprehensively described. It is very interesting and offers new insights into the MYC family.

Author Response

Thank you very much for your kind comments.

This manuscript is a resubmission of an earlier submission. The following is a list of the peer review reports and author responses from that submission.

Round 1

Reviewer 1 Report

I have very carefully checked each word, sentence, syntax, chapter organization, and individual reference papers cited within the pending manuscript. It took more than 4 hours not to make any careless mistake for me to complete this important task as an anonymous peer-reviewer. Here is the summary of my comments.

1) The title is not sharp enough. It is too broad and vague. I did not feel any enthusiasm and/or philosophy about the MYC research of the authors. MYC is a very important transcription factor not only in hematopoietic tumors but also solid spontaneous adult cancers. However, the content apparently includes large amount of information on blood cancer with little information about solid carcinoma. If so, the title should include the more indicative words, for example, "mainly focusing on hematopoietic tumor pathophysiology"

2) The authors have missed a number of critical MYC-related signaling mechanisms in this review article. The authors should have done their own homework to look for and collect these missing information. I am afraid that this is the biggest deficiency of this review manuscript.

3) There are a number of misleading sentences to explain the underlying mechanisms. For example, it is not true that the MYC MBII is able to bind to TRRAP. The authors should carefully read and investigate precisely how TRRAP was originally identified and purified (isolated). Believe or not, during peer-reviewing this manuscript, I carefully checked individual original paper's figure that were published many years ago (available via PubMed) in order not to make any careless mistake.

4)  Accordingly, I noticed that there are many inappropriate references cited. In a review article, the authors must cite the 'original' research papers.

5) In Fig. 1 and Fig. 4, there were no detailed figure legends.

6) The brief chapter index is needed at first.

7) The English written by the authors need to be definitely polished up by any editing service agency. 

Reviewer 2 Report

This review on MYC written by Scafuro and colleagues summarizes early literature on MYC. While the title suggests a focus on target gene regulation, the review does not quote recent literature on this topic, but focuses on reports published before whole genome approaches became available. A reader will not be able to get aware of the wealth of datasets published in the last 10 years and the heated discussion on whether MYC at all regulates specific target genes or is a general amplifier of gene expression. More recent work from the Amati, Eilers, Levens, Young and Wolf lab should be discussed (e.g. Lin et al, Nie et al, Walz et al, Sabo et al, Lorenzin et al). Also, the mechanism of target gene regulation seems outdated and the work does not discuss the possibility that MYC interacts and recruits directly transcription initiation and elongation factors such as CDK9 (Lin … Young, Cell) or SPT5/SPT6 (Baluapuri … Wolf, Molecular Cell).

Other major points:

1.: Figure 1 needs major revision. The overall Figure design suggests unique cellular functions of “physiological” and “oncogenic” Myc. But cellular phenotypes are then mostly given for physiological MYC. Many of them, such as angiogenesis, seem also important for tumorigenesis. The authors have to provide references for each cellular phenotype and their classification as physiological or oncogenic.

2.: Many sentences are very difficult to understand (e.g. “Aberrations affecting the MYC locus, polymorphisms in MYC regulatory sequences, and aberrant transduction pathways of MYC activation and repression are the main alterations that mutate protein-coding sequences“ and not precise. I recommend the authors to go through their complete paper and try to make their statements more precise.

3.: The authors first mention all MYC boxes and then only describe MB0 and MB1. The Figure on the other hand does not contain MB0. This section seems incomplete. The Interaction of Myc to other proteins seems arbitrary, how about AuroraA and WDR5 where even crystal structures exist?

4.:  Many statements need references, e.g. “Unlike MAX, MYC cannot homodimerize in vivo.”, “Its human cellular homolog, MYC, was identified in Burkitt lymphoma (BL) patients with t(8;14)(q24;q32), in which MYC is translocated to the IGH promoter and subsequently overexpressed.”, and the strange statement “Although MYC overexpression is associated with enhanced cellular proliferation, its inactivation or 122

downregulation leads quiescent cells to re-enter the cell cycle.”